# Subjective Experience and Perceived Benefits in Clients with Schizophrenia Following Participation in Metacognition Reflection and Insight Therapy (MERIT)

**DOI:** 10.3390/bs14060450

**Published:** 2024-05-27

**Authors:** Noa Tsuck-Ram, Adi Moka, Adi Lavi-Rotenberg, Libby Igra, Ilanit Hasson-Ohayon

**Affiliations:** 1Department of Psychology, Bar-Ilan University, Ramat Gan 5290002, Israel; noats5@mta.ac.il (N.T.-R.); mokaadi@biu.ac.il (A.M.); laviadi@biu.ac.il (A.L.-R.); lbi@psy.ku.dk (L.I.); 2Department of Community Mental Health, University of Haifa, Haifa 3498838, Israel; 3Department of Psychology, University of Copenhagen, DK-1165 Copenhagen, Denmark

**Keywords:** schizophrenia, metacognition, qualitative, MERIT, subjective

## Abstract

Schizophrenia spectrum disorders involve disturbances in the experience of the self, which are related to limited metacognitive ability. The aim of metacognition-based therapies is to improve metacognitive ability and, subsequently, self-management and recovery. Adding to the quantitative findings from a trial on the effectiveness of Metacognition Reflection and Insight Therapy (MERIT), in the current study, we report on a qualitative assessment of MERIT’s subjective perceived contribution. Twenty-seven patients with schizophrenia were interviewed after completing MERIT. Content analysis based on grounded theory was conducted by two independent raters. Most participants were satisfied with the therapy and reported improvement mainly in self-experience domains. The main contributors to perceived improvement pertained to the intervention process (e.g., therapeutic alliance and therapist interventions) as well as to the patient being an active agent of change. Perceived outcomes are particularly important among this cohort, as they often cope with limited metacognitive abilities, expressed by challenges in reflecting on themselves and others. The current study supports MERIT as a treatment that enhances positive outcomes for people with schizophrenia.

## 1. Introduction

In line with early discussions regarding schizophrenia [1,2], the emphasis of current views is on the major alterations of self that typify this disorder: for example, fragmentation, a limited sense of agency, and challenges in forming integrated and complex representations of self and others [3]. These challenges may result in deficits in self-management and decreased wellness [4] and have led to the development of treatment approaches aiming to enhance a coherent sense of self as an active and reflective agent [5].

A theoretical and empirically based metacognitive model for psychotherapy that addresses the abovementioned challenges among people with schizophrenia was recently developed and applied in several trials. Metacognitive Reflection and Insight Therapy (MERIT) [6] views metacognition as including a wide range of reflective processes, ranging from being aware of distinct experiences (e.g., identifying emotions) to synthesizing these experiences into a broader narrative [4,7]. The MERIT approach allows individuals to be aware of and reflect on their experiences and to perform an ongoing construction of integrative and holistic representations of the self and others; to make sense of one’s place in the world; to understand one’s own reactions to psychological difficulties; and to achieve a sense of agency in the world [6,8]. Metacognition, as viewed when practicing MERIT, refers to the conscious appraisal of one’s mental activities. These activities include reflecting on a continuum of discrete to complex experiences that may vary according to who is being reflected upon (e.g., self, other). Importantly, metacognition is considered multidimensional and includes four domains: self-reflectivity; understanding the other’s mind; decentration (reflecting on one’s place within a community); and mastery (one’s ability to adaptively use reflective knowledge) [8].

Applying MERIT, therapists adhere to eight core elements whose aim is to improve different aspects of metacognition [6]. These elements include focusing on the patient’s agenda; introducing the therapist’s mind; eliciting the patient’s narrative; focusing on a psychological problem; reflecting on interpersonal processes; reflecting on change; stimulating reflections on the self and others; and using this reflective knowledge adaptively [8]. Studies have shown MERIT to be feasible and acceptable in different psychiatric settings. Empirical evidence has been provided in three randomized controlled trials [9,10,11]; open trials [12,13]; session-by-session assessments of process and the contribution of specific core elements [14,15]; case study descriptions [16,17,18,19]; and two qualitative studies [20,21]. These studies have shown that MERIT promotes both clinical and subjective recovery.

Important findings emerged from the two abovementioned qualitative assessments. One, conducted by de Jong and colleagues [21], showed that 68% of the participants reported positive changes in how they viewed themselves and, in their ability, to conceptualize and cope with the challenges/difficulties in their lives. In addition, participants reported improvements in cognitive (64%), affective (57%), and interpersonal domains (43%). Participants attributed these changes to the therapeutic alliance, to their ability to express themselves, and to their assumption of an active role in treatment [21]. The results of the other qualitative assessment, conducted by Lysaker et al. [20], in which two therapy approaches were compared, showed that participants receiving either type of treatment—MERIT or supportive therapy—felt more confidence and self-esteem, could think more clearly about their lives, and were ultimately better able to set meaningful goals for themselves. However, the group receiving MERIT (as opposed to the supportive therapy group) reported being able to integrate their current experiences into the larger narratives of their lives. In addition, the MERIT group’s growth in self-esteem and cognitive clarity was qualitatively different from that of the group receiving supportive therapy: namely, the MERIT participants experienced an increased awareness of their thoughts and feelings and were able to place their challenges/difficulties within the context of a richer understanding of their histories [20].

Aiming to further explore the subjective experiences of participants receiving MERIT, and in line with the understanding that researchers must apply mixed methods when assessing the effectiveness of psychosocial interventions [22], we employed a qualitative approach in the current study. This qualitative assessment complements the quantitative assessment (i.e., the randomized controlled trial) that was conducted with the same sample [11], which showed significant interaction effects between group and time in both symptoms of schizophrenia and metacognition. As a further exploration of these promising results, we aimed in the current study to provide insights regarding the possible benefits and contributing factors of MERIT as perceived by clients. In line with the principles of qualitative methodology [23], the current study was exploratory in nature. We did not have explicit hypotheses but rather viewed participants’ spontaneous responses as potential sources of new and unexpected findings.

## 2. Method

### 2.1. Research Setting and Design

The current study was part of a randomized controlled trial in which the effectiveness of MERIT in psychiatric community settings in Israel was assessed (Clinicaltrial.gov ID NCT03427580). The quantitative findings of this trial are reported elsewhere [11]. The data were collected between the years 2017 and 2021. Approval for the study was obtained from the ethics committee of the Department of Psychology of Bar-Ilan University in Israel. Clients approached the clinic voluntarily seeking therapy for various psychological challenges and received a detailed explanation of the study; subsequently, they provided written consent and agreed to the use of their data for research. Following intake, the clients were randomized to one of two groups: the treatment group or the delayed-treatment control group. The data in the current study are based on individual face-to-face, semi-structured post-therapy interviews conducted with participants from both groups (both completed MERIT either at the beginning of the study or as a delayed condition).

### 2.2. Participants

A final sample of 27 participants took part in this qualitative study. All participants were diagnosed with schizophrenia spectrum disorders (i.e., schizophrenia or schizoaffective disorders), according to previous medical data and according to the Mini International Neuropsychiatric Interview (MINI) using the DSM diagnosis [24]. The exclusion criteria were intellectual disability, neurological disorders, active substance use disorder, acute psychosis, and risk of suicidal behavior, based on the intake interview. Participants were 18 years of age or older (mean age = 37.3, SD = 10.1, age range 23–56 years), and the majority were male (66.7%). In terms of marital status, the majority were single (81.5%), and the rest were either married (14.8%) or divorced (0.37%). The mean educational level was 12.4 years (SD = 1.76, min = 9 years, and max = 18). Fifty percent of the participants reported living with their parents (*n* = 13), 30.8% reported living alone (*n* = 8), 11.5% reported living in a hostel (*n* = 3), and 7.7% reported living in a supportive community (*n* = 2). Regarding employment, 51.9% reported being employed (*n* = 14), 22.2% reported supportive employment (*n* = 6), 18.5% reported being unemployed (*n* = 5), and 7.4% reported being a student (*n* = 2). All clients were receiving psychiatric medical treatment at outpatient clinics as well as rehabilitative services in the community.

### 2.3. Measures

#### 2.3.1. Qualitative Assessment

Narrative Evaluation of Intervention Interview (NEII) [25]. The NEII is a semi-structured interview aimed at evaluating the subjective experience of both processes and outcome of psychosocial interventions and psychotherapy. The interview lasted approximately half an hour and consisted of 16 open-ended questions formulated to encourage spontaneous reports of participants’ experiences. Sample questions were as follows: “What change, if any, took place during your participation in the intervention?” and “What was the intervention’s contribution to you and its subsequent impact? Please describe.” The NEII does not explicitly refer to any expected/specific outcome or change, and the aim of the questions was to explore how participants perceived, understood, experienced, and felt about the intervention. Studies in which the NEII has been used to explore participants’ perceived experience in psychotherapeutic settings, including MERIT, have shown high inter-rater reliabilities, e.g., [21,26,27]. In this study, Cohen’s Kappa coefficient was used to measure the degree of agreement (i.e., inter-rater reliability) [28] between the raters for the different themes. In the current study, inter-rater reliability for all of the categorical ratings was moderate to high, with Kappa coefficients ranging from 0.7 to absolute agreement.

#### 2.3.2. Qualitative Analysis

Interviews were obtained at the end of the trial by a psychologist. Content analysis was carried out in three stages, aligning with the grounded theory approach [29]. First, two judges (the first and second authors, who were trained in content analysis and did not take part in the interviewing or in providing therapy, to avoid biases) read the interviews to identify the degree of participants’ satisfaction with the therapy, the domains of change following the therapy, and the main factors leading to the perceived change. Note that themes identification was performed in line with the ideas of coding and meaning saturation of qualitative data [30]. During the second stage, the judges discussed and reached agreement on the coding themes that emerged from the interview data and developed rating anchors and a scoring system. The degree of satisfaction scores ranged as follows: 0 (not mentioned at all during the interview), 1 (negative reference), 2 (neutral reference), and 3 (positive reference). For the other main identified themes, the scoring was dichotomous: 0 (the theme was not mentioned at all during the interview) or 1 (the theme was mentioned during the interview). During the third stage, each judge read each participant’s entire interview and independently classified it according to the coding themes. Discrepancies in ratings were resolved through discussion until a consensus was reached. In addition to the presentation of themes, descriptive statistics are provided to show their frequencies.

### 2.4. MERIT Intervention

This approach to psychotherapy is operationalized as consisting of eight measurable elements that aim at promoting metacognition [6]. These elements that are mentioned in the introduction are as follows: focusing on the agenda of the client, introducing the therapist’s mind, reflecting on narrative episodes, exploring and conceptualizing psychological problems, reflecting on interpersonal processes, reflecting on therapeutic progress, stimulating the client’s self-reflectivity and understanding of others, and stimulating the client’s mastery, i.e., his or her ability to use reflective knowledge to cope with challenges [6]. This approach does not use a predetermined curriculum, and all elements are applied in a way that matches but does not exceed a client’s current metacognitive capacities. Therapists in the trial were 11 clinical or rehabilitation psychology interns who took part in a structured MERIT training prior to the trial. They participated in one weekly hour of group supervision provided by a MERIT expert (the last author of the current study). For detailed information on the intervention and trial, see the quantitative study of the trial [11].

## 3. Results

### 3.1. Level of Satisfaction

The distribution according to level of satisfaction is presented in Table 1. Most of the participants (77.8%, *n* = 21) were satisfied with the intervention. For example, one of the participants said the following: “*Among all the therapies I have been through, I think this therapy helped me the most. It didn’t help me to get completely out of my troubles, but I really appreciate the treatment I went through and I think it helped me.*” Another said, “*The treatment allowed me to be more self-confident; it strengthened my motivation and joy in life and my faith in people; and it also helped me function better.*” The rest of the participants were equally divided between those who were dissatisfied with the treatment and those who held a neutral position toward it. The participants who were dissatisfied (11.1%, *n* = 3) made statements such as “*I did not feel that participating in the treatment contributed to me or helped me; I talked about it a lot with my therapist*” or “*I felt that we talked about solutions in therapy, but in the end, nothing happened.*” Participants who held a neutral position (11.1%, *n* = 3) made statements such as “*I feel that there is a certain change, but something very general, something that started even before the treatment; I don’t know if it can be attributed to the treatment*” or “*I don’t know what or how the treatment helped; I believe that the change I feel has to do with the medications I take.*” Thus, these latter participants seemed to experience change but did not attribute it to their participation in MERIT.

### 3.2. Outcome Themes

Participants reported two main domains of change that they attributed to participating in MERIT: improvement in reflective abilities and improvement in self-management, as presented in Table 2.

Improvement in reflective abilities: Most participants mentioned improved reflective abilities (85.2%, *n* = 23). This improvement was evident with regard to self-awareness, including the ability to focus on the self, to develop a more coherent style of self-expression, to reflect on different processes and situations, and to gain a deeper understanding of these processes/situations. For example, “*I developed the ability to talk about topics that I had previously suppressed, and openness about topics that are hard to talk about.*” Improvement in reflective abilities was also evident with regard to being aware of the other. For example, one participant said, “*I learned that the other person could still love me despite everything… She [the therapist] wasn’t mad at me.*” Another participant said, “*She [the therapist] showed me another way of thinking; how to step into someone else’s shoes.*”

Self-management: The second domain in which change occurred and which was mentioned by participants (70.4%, *n* = 19), was self-management. This domain pertains to improved self-efficacy, confidence in one’s ability to manage symptoms and other stressors in life, and the promotion of personal goals. For example, one participant said, “*I made personal changes during the treatment; I got my driver’s license back, I joined the country club, I started being active and doing things that contributed to me in a good way.*” Another participant said, “*Following treatment there were changes. I am independent, I know how to ride the bus, where to get off and everything. Even at home I am more independent; I make my own soup!*” Another participant drew a connection between having a better sense of self and improved functioning: “*It [the therapy] gave me self-confidence, motivation, joy … it helped me to function better at home.*”

### 3.3. Process Themes

Participants perceived three related process factors as contributing to positive change: therapist interventions, therapeutic alliances, and client involvement.

#### 3.3.1. Therapist Interventions

This factor captures three main aspects of therapists’ therapeutic actions. Most participants described at least one of these aspects as being crucial for the intervention’s effectiveness (92.6%, *n* = 25): 1. Engaging in collaborative thinking (56%, *n* = 14), for example, “*I felt that my therapist was having a dialogue with me, like two intelligent people do. She allowed me to see a point of view that I hadn’t imagined before, and it was refreshing; I liked it and adopted it*” and “*The therapist asked questions, listened, and tried to understand more, with me, things I had always avoided*”; 2. Therapist self-disclosure (28%, *n* = 7), for example: “*She shared her personal opinion, as well as her professional opinion. When she shared her feelings in therapy, I felt that I understood more, I felt that it helped me*”; 3. Reflecting together about goals (40%, *n* = 10), for example, “*The therapist helped me see the problems differently and think outside the* box” and “*For me, the treatment was setting goals together and achieving them. One by one. Like for example the way we worked on my social anxiety, through conversations and exposure, until I felt there was a certain improvement.*”

Some participants (11%, *n* = 3) referred to certain therapist interventions in a negative way, such as, “I felt that the therapist was talking with me about solutions, but was not doing anything to help me actually realize those solutions.” One participant (3.7%) referred to therapist interventions in a neutral way: “It’s a bit hard for me to say what the therapist did. I’ve recently felt a kind of general change, but I find it difficult to attribute this change to something specific, to the treatment or to anything my therapist said/did.”

#### 3.3.2. Therapeutic Alliance

This factor captures the perceived participant–therapist collaboration and affective bond and was mentioned by 95.7% (n=26) of the participants. The most significant element in the therapeutic alliance, as mentioned by almost half of the participants (48.14%, *n* = 13), was the emotional (affective) bond between the two parties. For example, one participant said, “*For the first time a therapist remembered everything I said to him*” and “*[The therapist was] someone who talked to me as an equal and tried to understand me. Even though I was distant, he tried to understand me so I could speak freely with him.*”

In terms of the characteristics that contributed to the building of the therapeutic alliance, most participants (85.2%, *n*= 23) mentioned support, continuity, listening, honesty, and therapy as being a safe place to vent. Therapist empathy, in particular, was mentioned as a significant factor in building the therapeutic alliance; it was mentioned by 30% (*n* = 7) of the participants. For example, “*The therapist was empathic toward me, and helped me to be less judgmental toward myself and to better understand the situation*” and “*The therapist was empathic and very likable, and we laughed a lot together.*”

#### 3.3.3. Patient Involvement in Therapy

This factor describes participants’ actions during therapy. Most reported that they had taken such actions (92.6%, *n* = 25), mainly reporting three aspects. 1. Sharing. For example, “*It helped me to pour my heart out*”; “*The very fact that I talked about personal things that I almost never talked about in real life, skeletons from the closet, was already a blessing*”; “*I tried as much as possible to share emotions and thoughts*.” 2. Consistent attendance at meetings and commitment to the process. For example, “*I was very committed to the treatment, persistent, always on time. Even when it was raining, I didn’t miss a single session*”; “*I think I was very committed to the treatment because I always arrived, and on time, and I didn’t skip a single meeting. My therapist also told me she felt the same way [about my commitment]*”. 3. Working between sessions. All the participants mentioned how they continued to do the work between sessions: “*As part of the treatment, I have changed; my attitude has changed. When I leave the room, I feel a sense of a beginning and not an end. I keep thinking about the things that were said in the therapy, digesting them, trying to understand where everything comes from.*” (Table 3).

## 4. Discussion

Aiming to elaborate on the effectiveness of MERIT for people with schizophrenia, in the current study, we explored participants’ perceived experience in MERIT and the benefits that this therapeutic approach offered. As such, this study adds to the existing literature, reviewed in the Introduction, regarding the effectiveness, process, and subjective experience of MERIT. The vast majority of study participants (77.8%) reported being satisfied with the therapy, mainly in domains related to self-experience—that is, self-management and reflective abilities. In addition, participants perceived these changes as occurring due to the therapeutic alliance, therapist interventions, and the patient being an active agent of change.

The effectiveness of MERIT has been examined in a number of quantitative studies, including clinical trials [9,10,11,13,14] and case studies [4,16]. These studies revealed evidence supporting MERIT’s efficacy in improving metacognitive abilities, mainly in the domains of self-reflectivity and mastery. A previous qualitative evaluation of the effects of MERIT showed that participants found the intervention to be contributory mainly in the “cognition and reflection” domain and mentioned an increased understanding of themselves [21]. The contributing factors they identified were their own adherence/active role, their ability to “vent and self-express”, and the formation of an alliance with the therapist.

The current study strengthens the existing literature by presenting similar findings regarding the high level of satisfaction derived from the therapy, as reported by participants. It should be noted that our sample consisted only of participants who had completed the therapy, and, as such, the findings may be biased and should be considered with caution. In addition, it might be that participants’ perceived benefits can be attributed to their participation in therapy in general and not necessarily in MERIT. As the study took place partially during the COVID-19 pandemic, it might be that any therapy, at that time of experiencing general distress, would produce a positive outcome. That said, participants reported an improvement in the domains of reflective abilities and self-management, similar to the findings of de Jong et al. [21]. The current study also reveals similar process themes to those found earlier, including patient involvement in therapy, therapist interventions, and therapeutic alliance. However, the present study, by unveiling several sub-themes, broadens the understanding of how these process themes contribute to change.

Within the theme of patient involvement as a factor that contributes to change, participants specifically mentioned three key factors: sharing; committing to the process and attending meetings; and doing the therapeutic work between sessions. Such reports from patients are in line with findings from previous studies on psychotherapy in general suggesting that high treatment attendance is important for positive treatment outcomes, as it predicts therapy engagement, which in turn predicts treatment effectiveness [31,32,33]. These reports are also in line with the recovery literature, emphasizing the self as an active agent in one’s recovery process [34]. Thus, viewing the self as a source of positive change and growth is essential for one’s sense of self as an active agent in achieving meaningful goals [34].

Also, in the current study, participants discussed various therapist interventions that they felt contributed to the therapy, with an emphasis on the following: engaging in collaborative thinking, therapist self-disclosure, and reflecting together about goals. Collaborative thinking enables patient-therapist engagement, fostering a mutual exchange of ideas and cultivating the therapist’s commitment to the patient [35,36]. Furthermore, such collaborative thinking empowers the patient, as it fosters a heightened sense of competence and active participation throughout the therapeutic process [36]. Another therapist intervention deemed important for change (by the participants) was therapist self-disclosure. Consistent with prior research, these findings align with previous findings regarding the therapeutic benefits of therapist self-disclosure. Such benefits include enhancements of mental health, patient insight, and the overall improvement of the therapeutic alliance [37,38], which itself emerged as a central theme in the current study (i.e., as a contributing factor to change). Self-disclosure in the context of MERIT for patients with schizophrenia is explicitly encouraged and is one of its core elements, namely, introducing the therapist’s mind. This element can be especially contributory in the treatment of people who have schizophrenia in two ways. First, when the therapist invites the patient to enter an inter-subjective space and models their thought processes (e.g., speculating and even expressing doubt), the patient may be able to practice and learn more flexible and effective thought processes. Second, self-disclosure can convey to the patient a message of respect, inclusion, and emotional closeness. These aspects might be particularly important for patients with schizophrenia, given that society in general and mental health services in particular often convey messages of social distance, inferiority, and discrimination [39]. Two studies that assessed session-by-session changes in MERIT showed this element to be important in relation to improvement in emotional experience, therapeutic alliance, and general outcome [14,15].

The association between therapeutic alliances and favorable psychotherapy outcomes has been established across varied therapeutic approaches and client populations [40]. In regard specifically to people with schizophrenia, a recent meta-analysis revealed an association between strong therapeutic alliance and better treatment engagement and symptom improvement for this cohort [41]. Additionally, and consistent with our findings, therapists’ empathy, genuineness, and trustworthiness have been found to predict better reports of therapeutic alliance by clients with schizophrenia [42]. One possibility that emerges in the current study (i.e., in terms of participants’ experience of therapists’ empathy, listening, support, and honesty) is that the MERIT therapeutic position helps therapists overcome some of the difficulties that may arise in establishing empathy and a therapeutic alliance with clients who have schizophrenia [6]. Going forward, researchers exploring therapists’ subjective experience with MERIT could examine the possibility that applying MERIT contributes to an empathic stance.

Developing a sense of agency was also identified as a contributory factor to MERIT’s effectiveness. As mentioned previously, when discussing perceived outcomes, developing a sense of agency stands as another crucial core element in MERIT, potentially augmenting patient engagement in therapy and consequently promoting positive outcomes [43]. Through active participation in sessions, and by actively strengthening the therapeutic alliance, individuals diagnosed with schizophrenia play an active role in their therapeutic journey, facilitating the development of a sense of agency that, in turn, may influence their subjective sense of competence in the world [4,33,42].

The current study stands as a complementary study to the quantitative assessment of the same trial reported elsewhere [11]. Whereas the focus of the quantitative study was on metacognition and symptom improvement, the focus of the current qualitative study is on perceived processes and outcomes, which are not necessarily related to the primary examined quantitative outcomes. The quantitative assessment showed improvement in metacognition, including in the mastery domain, which highlights a sense of agency (i.e., one’s ability to use reflective knowledge to cope with different challenges). It also showed symptomatic improvement, or at least maintenance, among the study group, whereas the control group showed deterioration. Taken together, the quantitative and qualitative findings show MERIT to be an effective therapy for increasing one’s sense of agency. The quantitative assessment supported this finding by showing improvement in metacognition’s mastery domain, and the qualitative assessment did so by showing perceived improvement in reflection and management, suggesting that both together assist one in managing life better and being more reflective. In addition, one of the process factors that participants viewed as contributing to a positive outcome was their involvement, further supporting the importance of enhancing agency via MERIT. Interestingly, participants in this qualitative study did not mention symptoms, suggesting that either they did not perceive the therapy as targeting the reduction of symptoms or they did not perceive any symptomatic change.

The current study had some limitations. A few of the interviews produced a relatively limited amount of data, as these participants responded in a laconic way. In addition, we collected the data only once, after therapy completion. We did not do a longitudinal follow-up measurement, and therefore information regarding the maintenance of the therapy’s effects is lacking. In addition, we examined the effects of MERIT without comparing its effectiveness to other therapeutic approaches. Such a comparison would have further validated the uniqueness of MERIT. Lastly, in the current qualitative investigation, two out of the five authors were therapists in the trial sessions, and one was a supervisor; the other two authors, who were not part of the trial, functioned as independent judges of the data. In this way, the researchers’ potential biases were reduced, and greater open-mindedness to new and/or unexpected content was ensured.

With these limitations in mind, the current study emphasizes the importance of exploring participants’ subjective experiences so as to provide a rich and comprehensive understanding of an intervention’s effects. In this study, we outlined MERIT’s processes and outcomes, as perceived by study participants, underlining the importance of specific interventions, therapeutic alliances, and the client being an active agent. Further research should validate these findings longitudinally and in relation to other types of psychotherapies. In conclusion, the effectiveness of the MERIT approach has been further established in this study. Specifically, MERIT seems to be an effective therapy for people with schizophrenia who have challenges in self-experience and metacognition.

## Figures and Tables

**Table 1 behavsci-14-00450-t001:** Level of perceived satisfaction with treatment.

LEVEL OF SATISFACTION		
Not Satisfied	Neutral	Satisfied
11.10%	11.10%	77.80%

**Table 2 behavsci-14-00450-t002:** Perceived outcome in self-management and reflective abilities.

OUTCOME THEMES		
Self-management	Reflective Abilities	
70.40%	85.20%	Mentioned This Domain
29.60%	14.80%	Didn’t Mentioned This Domain

**Table 3 behavsci-14-00450-t003:** Process themes: therapist interventions, therapeutic alliance, and patient involvement.

PROCESS THEMES		
Therapist Interventions	Therapeutic Alliance	Patient Involvement
92.6%	95.7%	92.6%

## Data Availability

The datasets presented in this article are qualitative and based on interviews that include personal information. Therefore, in the interest of ensuring participants’ confidentiality, the data are not readily available.

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
