# Peer review of "Subjective Experience and Perceived Benefits in Clients with Schizophrenia Following Participation in Metacognition Reflection and Insight Therapy (MERIT)"

_behavsci, 2024, doi:10.3390/bs14060450_

Round 1
Reviewer 1 Report (Previous Reviewer 3)
Comments and Suggestions for Authors
I have reviewed the manuscript for the second time and regret to note that there has been no discernible improvement in the quality of the work. Despite the feedback provided during the initial review, the manuscript still exhibits significant deficiencies in its methodology, analysis, and reporting. It is imperative that the authors address the issues raised in both reviews comprehensively to ensure the scientific rigor and validity of the study findings.
Sentence Structure:
Some sentences are overly complex and could be simplified for clarity. For example, the sentence "The remaining participants were evenly divided between those who were dissatisfied with the treatment and those who held a neutral position toward it" could be rewritten as "The rest of the participants were evenly split between those dissatisfied with the treatment and those who remained neutral."
Clarity of Findings:
While the results are presented, the narrative could benefit from clearer explanations of how the findings contribute to the existing literature or theoretical frameworks. The discussion section should provide a comprehensive analysis of the results and their implications, including any unexpected findings or limitations.
Flow of Information:
The narrative should flow logically from one section to the next, guiding the reader through the research process and findings. Ensure that each section transitions smoothly into the next and that all information presented is relevant to the research question.
Avoiding Overgeneralization:
The interpretation of the results should be cautious and avoid overgeneralization. While the findings may suggest certain trends or patterns, it is essential to acknowledge the limitations of the study and the potential for alternative explanations.
Consideration of Alternative Explanations: The authors should consider alternative explanations for the observed results and discuss them in the interpretation section. This could include factors such as participant characteristics, external influences, or methodological limitations that may have affected the outcomes.
Ethical Considerations:
It is essential to provide information on ethical approval and informed consent procedures in the methodology section. This ensures transparency and demonstrates adherence to ethical guidelines in research involving human participants.
Replicability:
The methodology should provide sufficient detail to allow other researchers to replicate the study if desired. This includes information on participant recruitment, intervention procedures, outcome measures, and data analysis techniques.
Sampling Methodology:
The methodology lacks clarity regarding the sampling technique employed. Without a clear description of how participants were selected, it is challenging to assess the representativeness of the sample and the generalizability of the findings.
Sample Size and Diversity:
The sample size seems relatively small, with only 27 participants. A larger sample size would enhance the generalizability of the findings. Additionally, it would be beneficial to include more diversity in the sample to ensure that the results are applicable to a wider range of individuals.
Sample Size Determination:
The study does not provide justification for the sample size chosen. A power analysis or rationale for the sample size calculation is essential to ensure that the study is adequately powered to detect meaningful effects.
Participant Recruitment:
The methodology does not specify how participants were recruited into the study. Recruitment methods can influence the composition of the sample and introduce biases that may impact the validity of the results.
Ethical Considerations:
The manuscript lacks information on ethical considerations, such as informed consent procedures, confidentiality assurances, and institutional review board approval. Ethical oversight is crucial in research involving human participants to protect their rights and welfare.
Intervention Procedures:
While the study evaluates the effectiveness of a specific intervention, the methodology provides limited detail on the intervention procedures. A comprehensive description of the intervention protocol, including its components, duration, frequency, and delivery format, is necessary for replication and interpretation of the results.
Control Group:
The absence of a control group is a notable deficiency in the methodology. Without a control group for comparison, it is challenging to attribute observed changes solely to the intervention, as other factors may have influenced the outcomes.
Outcome Measures:
The study lacks clarity regarding the outcome measures used to assess the effectiveness of the intervention. Clear operational definitions of outcome variables and valid, reliable measurement instruments are essential for ensuring the validity and reliability of study findings.
Data Analysis:
The methodology provides limited information on the statistical analyses employed. Transparency regarding data analysis procedures, including statistical tests used, adjustments for multiple comparisons, and handling of missing data, is necessary for evaluating the robustness of the results.
Comments on the Quality of English Language
The manuscript demonstrates persistent issues with the clarity and coherence of the English language. Syntax errors, awkward phrasing, and grammatical inconsistencies are prevalent throughout the text, impeding comprehension and detracting from the professionalism of the work. It is essential for the authors to seek assistance from a proficient English language editor to rectify these linguistic shortcomings and enhance the readability of the manuscript.
Author Response
Responses to Reviewer #1 comments:
- Some sentences are overly complex and could be simplified for clarity. For example, the sentence "The remaining participants were evenly divided between those who were dissatisfied with the treatment and those who held a neutral position toward it" could be rewritten as "The rest of the participants were evenly split between those dissatisfied with the treatment and those who remained neutral."
Authors: Thank you for this comment. We have edited the entire paper with the assistance of a professional English-language editor.
- While the results are presented, the narrative could benefit from clearer explanations of how the findings contribute to the existing literature or theoretical frameworks. The discussion section should provide a comprehensive analysis of the results and their implications, including any unexpected findings or limitations.
Authors: We have edited the Discussion section to better relate to the existing literature on both recovery and psychotherapy.
- The narrative should flow logically from one section to the next, guiding the reader through the research process and findings. Ensure that each section transitions smoothly into the next and that all information presented is relevant to the research question.
Authors: We have edited the manuscript accordingly.
- Avoiding Overgeneralization: The interpretation of the results should be cautious and avoid overgeneralization. While the findings may suggest certain trends or patterns, it is essential to acknowledge the limitations of the study and the potential for alternative explanations.
Authors: In the revised version we edited the discussion to highlight limitations both at the end of the discussion in a limitation paragraph and earlier in the discussion when mentioning possible biases and alternative explanation (e.g. that responses were biased due to not including participants who did not complete therapy).
- Consideration of Alternative Explanations: The authors should consider alternative explanations for the observed results and discuss them in the interpretation section. This could include factors such as participant characteristics, external influences, or methodological limitations that may have affected the outcomes.
Authors: This was added to discussion- e.g. mentioning the context of covid-19.
- Ethical Considerations: It is essential to provide information on ethical approval and informed consent procedures in the methodology section. This ensures transparency and demonstrates adherence to ethical guidelines in research involving human participants.
Authors: This data is available in method section – including obtaining ethical approval and informed consent procedure.
- Replicability: The methodology should provide sufficient detail to allow other researchers to replicate the study if desired. This includes information on participant recruitment, intervention procedures, outcome measures, and data analysis techniques. Sampling
Authors: This data is available in method.
- Methodology: The methodology lacks clarity regarding the sampling technique employed. Without a clear description of how participants were selected, it is challenging to assess the representativeness of the sample and the generalizability of the findings.
Authors: Data is available in method. The study is part of a large scale one which we mentioned and added the relevant reference. We mention that clients were self-referred to clinic, their assessment procedure etc. please see setting and participants sections.
- Sample Size and Diversity: The sample size seems relatively small, with only 27 participants. A larger sample size would enhance the generalizability of the findings. Additionally, it would be beneficial to include more diversity in the sample to ensure that the results are applicable to a wider range of individuals.
Authors: This N is acceptable in qualitative studies. This is not quantitative analysis- and only descriptive statics is provided. We cannot include more diversity as the trail completed.
- Sample Size Determination: The study does not provide justification for the sample size chosen. A power analysis or rationale for the sample size calculation is essential to ensure that the study is adequately powered to detect meaningful effects.
Authors: In qualitative assessment power analysis in not appropriate. Rather, arriving to saturation in themes identification is the applied process. To make this clear we added this explanation to method.
- Participant Recruitment: 3 The methodology does not specify how participants were recruited into the study. Recruitment methods can influence the composition of the sample and introduce biases that may impact the validity of the results.
Authors: Sees response to comment # 13.
- Ethical Considerations: The manuscript lacks information on ethical considerations, such as informed consent procedures, confidentiality assurances, and institutional review board approval. Ethical oversight is crucial in research involving human participants to protect their rights and welfare.
Authors: See response to comment # 11. This data is available we also send the journal the approval as requested.
- Intervention Procedures: While the study evaluates the effectiveness of a specific intervention, the methodology provides limited detail on the intervention procedures. A comprehensive description of the intervention protocol, including its components, duration, frequency, and delivery format, is necessary for replication and interpretation of the results.
Authors: We added this to the last section on method. We elaborated on the MERIT approach and elements.
- Control Group: The absence of a control group is a notable deficiency in the methodology. Without a control group for comparison, it is challenging to attribute observed changes solely to the intervention, as other factors may have influenced the outcomes.
Authors: This is mentioned in limitations section. We wrote that we did not compare MERIT to another approach to treatment and discussed this.
- Outcome Measures: The study lacks clarity regarding the outcome measures used to assess the effectiveness of the intervention. Clear operational definitions of outcome variables and valid, reliable measurement instruments are essential for ensuring the validity and reliability of study findings.
Authors: This is a qualitative study and no outcome measures were used. Rather, open ended interview was used on uncover subjective perceived processes and outcome. This is explained in method.
- Data Analysis: The methodology provides limited information on the statistical analyses employed. Transparency regarding data analysis procedures, including statistical tests used, adjustments for multiple comparisons, and handling of missing data, is necessary for evaluating the robustness of the results.
Authors: This is a qualitative study and the only statics that is provided is descriptive as mentioned in manuscripts. No statistical tests were applied analyzing the qualitative data.
Reviewer 2 Report (New Reviewer)
Comments and Suggestions for Authors
Psychotherapy should be an important part of therapy in patients with schizophrenia and related disorders, so I was interested in reading the article. (I have long treated with medication, of course, as well as CBT.)
The article is fine to publish but I believe some of my comments may help to create a stronger article:
-You should give a sentence in the Intro about what are exactly the scz-related disorders. Also, given this article is about metacognition, you *really* have to discuss metacognition more. The few lines (e.g. line 46 approx) are not sufficient.
-line 38 above <space> mentioned
-line 380 .... was "ensured"
-Figures 1 and 2 do not add any useful information (and are blurry too). You are comparing three numbers that I think a little table would be fine -- graphs really should be used for other purposes.
- Line 171: "The majority of the participants (77.8%, n=21) were satisfied with the intervention... based on participants' statements."
- I know it is very hard to do proper research in the field of therapy, so I feel there is material to be learned about your work. However, what do these results really mean? This is something for you to think about. The 77.8% figure I guess gives some information and is ok to be published, but still, how valid is it really? Something for you to think about and perhaps write a little bit more about. (Also, perhaps it is wrong to put the three decimal figure in the Abstract -- perhaps more honest just to say "majority of participants....")
Author Response
Responses to Reviewer #2 comments:
- You should give a sentence in the Intro about what are exactly the scz-related disorders. Also, given this article is about metacognition, you *really* have to discuss metacognition more. The few lines (e.g. line 46 approx) are not sufficient.
Authors: In response to this point, we have added the relevant requested information on metacognition to the Introduction. We added data regarding the schizophrenia-related disorders to the Method section.
- line 38 above <space> mentioned
Authors: Changed.
- line 380 .... was "ensured"
Authors: Changed.
- Figures 1 and 2 do not add any useful information (and are blurry too). You are comparing three numbers that I think a little table would be fine -- graphs really should be used for other purposes.
Authors: We deleted the figures and added tables instead. We did the same with regard to Figure 3 as well.
- Line 171: "The majority of the participants (77.8%, n=21) were satisfied with the intervention... based on participants' statements - I know it is very hard to do proper research in the field of therapy, so I feel there is material to be learned about your work. However, what do these results really mean? This is something for you to think about. The 77.8% figure I guess gives some information and is ok to be published, but still, how valid is it really? Something for you to think about and perhaps write a little bit more about. (Also, perhaps it is wrong to put the three decimal figure in the Abstract -- perhaps more honest just to say "majority of participants....")
Authors: In response to this point, we have deleted the number in the Abstract and edited the Discussion to include the limitations of qualitative data when drawing conclusions.
Round 2
Reviewer 2 Report (New Reviewer)
Comments and Suggestions for Authors
Thank you for making these changes.
This manuscript is a resubmission of an earlier submission. The following is a list of the peer review reports and author responses from that submission.
Round 1
Reviewer 1 Report
Comments and Suggestions for Authors
TITLE
Instead of “clients” I would recommend “patients”.
AFFILIATIONS
Please, provide city for AFFILIATIONS number 2 and 3.
CORRESPONDENCE
Please, delete the gmail address. One address is enough, and I would recommend to rather keep the one from the official affiliation.
KEYWORDS
I would delete “qualitative” and “subjective”. I would recommend using not only the acronym but the full description of the instrument used. The best combination of Keyword would be, in my opinion: Schizophrenia; MERIT; Metacognition: Reflection; Insight; Therapy.
INTRODUCTION
Schizophrenia is not an illness. Please delete all “illness”, “disease”, “sickness” from the article. Schizophrenia is a syndrome in real life that was artificially condensed in ICD-11 and DSM-5 as a disorder. The Authors should use disorder or syndrome, but never “illness”, “disease”, or “sickness”.
For a matter of coherence I would recommend to use capital letters for all the four domains in MERIT (lines 46-49): Self-Reflectivity; Understanding the Other’s Mind; Decentration; and Mastery.
I believe there is no need for commas in “vent” at (line 67).
Please, provide reference(s) for sentences at the paragraph from line 68 to line to 78.
Please, explain all acronyms in the manuscript, at least once, when presented for the first time, eg “RCT” (line 83).
Some authors believe that the self is just an illusion created by the brain. Therefore there are researchers and clinicians that affirm that people with schizophrenia lack the grasp of their own self. Once without sense of self all psychopathology and phenomenology in schizophrenia is much easier to explain through a lot of mechanisms, eg autism, aberrant salience, double bookkeeping, jumping to conclusions, hallucinations, delusions, etc. I believe the Authors would like to read, at least one paragraph, regarding the opinion of Authors in such topic.
METHODS
Please, explain all acronyms in the manuscript, at least once, when presented for the first time, eg “IDNCT” (line 94), “SD” (line 109).
Please, avoid italics for English words (from line 139 to 142). I would recommend the use of italics only for non-English words.
The Authors should clarify if they used ICD-11 or DSM-5 for diagnosis of schizophrenia.
The Authors should quantify the amount of anti-psychotics used in the patients, eg applying the chlorpromazine equivalents (CPZE). That is a very important variable to take in account in patients with schizophrenia.
All patients need to be submitted, at least once, to a complete diagnostic march in order to exclude organic psychosis. Authors should not be hostages of the Cartesian catastrophe nor the Freudian fraud.
Please, provide blood work to exclude HIV, syphilis, metabolic, hormonal dysfunctions; urinalysis to exclude the abuse of drugs; brain Magnetic Resonance Imaging (MRI) to exclude encephalic lesion; Electroencephalogram (EEG) to exclude temporal lobe epilepsy; lumbar puncture (LP) to exclude encephalitis; neuropsychological assessment to exclude dementia, mental retardation, personality disorder, etc.
For more than 100 years that research with patients with schizophrenia mixed primary (idiopathic) schizophrenia with secondary (organic) schizophrenia. That mistake condemned psychiatry to medieval times, and should be avoided in future studies. Otherwise we will be just repeating ad nauseam the same mistake. Please, beware of pseudo-schizophrenia!
RESULTS AND DISCUSSION
These sections should be re-written according to the new diagnoses given to all patients. I would recommend a comparison of primary schizophrenia vs secondary schizophrenia patients. Sample should be increase for more robust statistics.
FIGURES
Figures have low quality. Please provide new ones. Avoid colors for economic and ecological reasons. I also would not recommend two decimal numbers for percentages. One seems to be enough.
REFERENCES
Kraepelin and Bleuler’s work have only historical interest. Both Authors’ definitions of schizophrenia are today completely outdated and obsolete. The same for other classic Authors such as Leonhard, Conrad or Schneider, etc. Please provide better and more recent bibliography regarding schizophrenia.
References at number 15 and 20, and all of them after number 26 are not according to the Journal instructions to Authors. Please review it carefully, and create a new Bibliography,
DISCLAIMER
After the section of REFERENCES there is an empty section. I can’t understand why.
Reviewer 2 Report
Comments and Suggestions for Authors
I thought the paper was clear and original. It's well-written and straightforward. I do not have any major suggestions for improvement.
I noticed a couple of typos in the manuscript:
1) On page 2, I wasn't sure why only two of the four domains were capitalized (self-reflectivity, etc.).
2) I think there is another typo after the phrase "specific core elements (1419)
Reviewer 3 Report
Comments and Suggestions for Authors
Methodological Rigor:
The study lacks a clear explanation of the sampling strategy. The method of participant selection and the criteria for inclusion/exclusion are not thoroughly detailed. This raises concerns about the generalizability of the findings.
Participant Demographics:
The study mentions that 27 participants were interviewed, but there is limited information about the demographic characteristics of the sample beyond age and gender. Providing a more comprehensive overview, including socio-economic status and educational background, would enhance the study's applicability to a broader population.
Qualitative Assessment Tool:
The Narrative Evaluation of Intervention Interview (NEII) is used as the qualitative assessment tool. While it is briefly described, more information is needed about the tool's reliability, validity, and any potential biases. Additionally, details on how the questions were formulated and whether they underwent pre-testing would strengthen the study.
Data Collection Timing:
The data were collected once, after therapy completion. A lack of follow-up interviews or assessments limits the study's ability to explore the sustainability of the reported improvements over time. Longitudinal data would provide a more robust understanding of the therapy's lasting effects.
Limited Comparison:
The study focuses on the positive aspects of MERIT without comparing its effectiveness to other therapeutic approaches. A comparative analysis with alternative treatments or control groups would strengthen the evidence for the unique contribution of MERIT.
Statistical Analysis of Qualitative Data:
The study lacks a detailed discussion on the statistical analysis applied to qualitative data. While the qualitative findings are presented descriptively, a more explicit discussion on the chosen methods, inter-rater reliability, and potential biases would enhance the credibility of the results.
Integration of Qualitative and Quantitative Data:
The study mentions a quantitative assessment (RCT) conducted with the same sample, but there is limited integration of qualitative and quantitative findings. A more thorough discussion on how these data complement each other or provide a comprehensive understanding of the intervention's effectiveness is necessary.
Transparency and Reflexivity:
The study lacks a discussion on reflexivity, acknowledging the researchers' potential biases and the impact of their perspectives on data interpretation. Including such reflections enhances the transparency and credibility of the research.
By addressing these considerations, the study could strengthen its methodological rigor, broaden the applicability of its findings, and provide a more comprehensive understanding of the therapeutic processes and outcomes associated with MERIT.
Comments on the Quality of English Language
The overall grammar and language usage in the text appear to be clear and well-structured. Sentences are mostly grammatically correct, and the language used is appropriate for a scientific context.
